# NIR Indocyanine–White Light Overlay Visualization for Neuro-Oto-Vascular Preservation During Anterior Transpetrosal Approaches: A Technical Note

**DOI:** 10.3390/jcm14196954

**Published:** 2025-10-01

**Authors:** Leonardo Tariciotti, Alejandra Rodas, Erion De Andrade, Juan Manuel Revuelta Barbero, Youssef M. Zohdy, Roberto Soriano, Jackson R. Vuncannon, Justin Maldonado, Samir Lohana, Francesco DiMeco, Tomas Garzon-Muvdi, Camilo Reyes, C. Arturo Solares, Gustavo Pradilla

**Affiliations:** 1Department of Neurosurgery, Emory University, Atlanta, GA 30303, USAgpradil@emory.edu (G.P.); 2Department of Oncology and Hemato-Oncology, University of Milan, 20122 Milan, Italy; 3Department of Otorhinolaryngology-Head and Neck Surgery, Emory University, Atlanta, GA 30308, USA; 4Department of Neurosurgery, Fondazione IRCCS Istituto Neurologico Carlo Besta, 20133 Milan, Italy; 5Department of Neurosurgery, Johns Hopkins University, Baltimore, MD 21287, USA

**Keywords:** ICG, near-infrared, neurovascular preservation, cochlear localization, MDK rhomboid, anterior petrosectomy, pretemporal approach

## Abstract

**Objectives**: Anterior petrosectomy is a challenging neurosurgical procedure requiring precise identification and preservation of multiple critical structures. This technical note explores the feasibility of using real-time near-infrared indocyanine green (NIR-ICG) fluorescence with white light overlay to enhance visualization of the petrous internal carotid artery (ICA) during transpetrosal drilling. We aimed to assess its utility for planning and performing modified Dolenc–Kawase drilling. **Methods**: We integrated NIR-ICG and white light overlay using a robotic microscope with simultaneous visualization capabilities. This technique was applied to improve neurovascular preservation and skull base landmark identification. Intraoperative video frames and images were captured during an anterior transpetrosal approach for a petroclival meningioma, with technical details, surgical time, and feedback documented. **Results**: Real-time NIR-ICG with white light overlay successfully identified the posterior genu, horizontal petrosal segment, anterior genu, and superior petrosal sinus. It facilitated precise localization of cochlear landmarks, enabling tailored drilling of the Dolenc–Kawase rhomboid according to patient anatomy and accommodating potential anatomical variants. **Conclusions**: This approach could enhance intraoperative safety and improve exposure, possibly reducing neurovascular risks without extending operative time. It may serve as a valuable adjunct for complex skull base surgeries.

## 1. Introduction

The extradural pretemporal anterior transpetrosal approach (ATPA) provides access to critical skull base regions, treating conditions like trigeminal and vestibular schwannomas, chordomas, and petroclival meningiomas [1].

Over time, the ATPA corridor has expanded through several surgical refinements: the “extended Kawase” technique has been introduced, extending beyond the inferior petrosal sinus toward the jugular tubercle to increase exposure further [2]. Parallel innovations, like the modified Dolenc–Kawase (MDK) approach, combine classical petrous apex drilling with Dolenc-style interdural cavernous sinus exposure (and anterior clinoidectomy), optimizing access to spheno-cavernous and petroclival lesions [3,4]. Other hybrid strategies incorporate Hakuba’s orbitozygomatic interdural window for Meckel’s cave access (“Hakuba–Dolenc–Kawase approach”), and the so-called ATSTA merges ATPA with subtemporal/transcavernous extension—strategies intended to enlarge the operative corridor but often at the cost of increased complexity [5,6]. Despite these advancements, challenges remain due to narrow operative corridors and anatomical constraints, with reported complications including facial palsy (6.2%) and hearing loss (3.9%) [5,6,7,8].

Effective use of ATPA hinges on identifying and preserving neurovascular structures, including the petrous ICA, geniculate ganglion, cochlea, facial nerve, and internal acoustic canal. However, anatomical variability limits the consistency of established osseous and dural landmarks, particularly when defining and optimizing the MDK rhomboid’s window. Furthermore, many traditional landmarks were developed for vascular routes that require direct ICA exposure—an objective that is rarely necessary in contemporary skull-base oncology—leaving surgeons to infer the ICA’s position indirectly.

Despite technical refinements and comparative anatomical work, drilling in proximity to the petrous ICA and cochlea still relies heavily on expert subjective orientation and broad bony landmarks. This dependence renders intraoperative localization vulnerable to operator-dependent and systematic errors. We therefore propose integrating near-infrared indocyanine-green (NIR-ICG) visualization with a digital white-light overlay to provide real-time, extradural, objective guidance to the petrous ICA. Our working hypothesis is that ICG-assisted ATPA/MDK might improve the landmarks identification and tailored posteromedial rhomboid drilling while preserving the ICA, cochlea, and IAC content integrity. This study will evaluate the feasibility and workflow impact of the technique compared with standard landmark-based practice.

## 2. Materials and Methods

### 2.1. Anatomical Report

One embalmed, un-injected adult cadaveric specimen was used. A 0.6 mm axial multi-slice helical CT scan was performed pre-procedure. Stereotaxic coordinates were recorded with a 1 mm registration tolerance (Stryker System II). The pulsatile flow was simulated using a customized perfusion device (unpublished data), and indocyanine green (ICG, 50 mg/100 mL) was administered during the procedure. Stepwise dissection replicated the clinical scenario, emphasizing anatomical landmarks. The trial adhered to institutional protocols, ethical guidelines for cadaveric research, and the “CACTUS” standards. Consent was obtained for scientific use and illustrative purposes [9].

### 2.2. Clinical Case

A prospective single-case feasibility study was designed to evaluate the role of intraoperative indocyanine green videoangiography (ICGva) with near-infrared (NIR) white-light overlay in enhancing anatomical delineation during the anterior transpetrosal approach (ATPA). The technique aimed to facilitate real-time visualization of the posterior genu, horizontal segment, and anterior genu of the petrous internal carotid artery (ICA), thereby improving the definition of skull base landmarks and enabling a more precise extension of the modified Dolenc–Kawase (MDK) rhomboid. Pre- and post-operative clinical records and imaging data were reviewed. Postoperative CT scans were analyzed, and in-hospital charts and MRI were assessed for complications.

The selected case involved a patient diagnosed with an extra-axial lesion of the posterior fossa, for whom ATPA was indicated due to the tumor’s proximity to the ventrolateral brainstem and the petrous ridge’s superior rim. High-resolution preoperative imaging demonstrated a petrous temporal bone thickness of less than 1 cm at the level of the posterior middle cranial fossa (MCF) floor above the horizontal carotid canal. Preoperative imaging excluded any dehiscence of the carotid canal, which was considered a contraindication for this adjunct technique.

Preoperative MRI and postoperative CT scans were co-registered, and, by manual (ICA and tumor) and semi-manual (skull) segmentation, a 3D model was created (3Dslicer, version 5.6.2) to illustrate the spatial relationship of fluorescent findings and the proposed landmarks.

### 2.3. Instrumentation

A ZEISS KINEVO 900^®^ robotic microscope (Carl Zeiss, Oberkochen, Germany) with ICGva modality was used for both experiments. The ZEISS INFRARED800 mode digitally overlaid ICG fluorescence (700–850 nm, max 805 nm) with white light, potentially eliminating workflow interruptions seen in prior devices. Standard microsurgical instruments were employed. A 12.5 mg bolus of ICG diluted in 10 mL saline was administered intraoperatively at the surgeon’s discretion. Standard microdissection techniques and neurophysiological monitoring were used to preserve the fourth, fifth, sixth, and seventh-eighth cranial nerves. Ethical approval was obtained (IRB 00000332; Approval: 6 May 2020), and the study adhered to the Declaration of Helsinki principles. The patient provided informed consent.

### 2.4. Anatomical Landmarks

The following landmarks were sequentially used for the scope of our study:Cochlear safety line (CSL): Defined as the perpendicular line connecting the lateral rim of the foramen ovale to the transition fold between the roof and anterior wall of the internal acoustic canal (IAC). This line laterally delineates the cochlea [10].Petrous ICA (pICA): Identified using intraoperative ICGva.Cochlear line (CL): Projected perpendicularly from the petrous ICA–GSPN intersection (measured during ICGva) onto the IAC. This line marks the lateral position of the basal cochlear turn [11].Carotido-cochlear distance (CCD): Adapted from Dew et al., measured as the distance between the medial ICA wall (defined with ICGva) and the geniculate ganglion at the fallopian hiatus [12]. Probing through the hiatus or GSPN electrical stimulation with facial nerve EMG confirmation refined this measurement. A 7 mm distance from the geniculate ganglion crotch was designated the safest boundary for identifying the basal cochlear turn [12].

## 3. Results

### 3.1. Cadaveric Simulation

A non-injected embalmed cadaveric head was cannulated at the common carotid arteries bilaterally, with the external carotid arteries and internal jugular veins clipped. The cannulas were connected to an in-house perfusion system to simulate blood circulation and alternating pulsation. An extended middle fossa approach with pretemporal-transcavernous and anterior transpetrosal techniques was performed as previously described [4,13]. After flattening the temporal squama to the middle fossa floor level, the temporal dura propria was peeled, and the middle meningeal artery (MMA) transected. Dissection began at the foramen ovale (FO), separating the dura propria from the trigeminal epineural sheaths and Meckel’s cave’s dorsolateral wall. Dissection continued medially, exposing the greater superficial petrosal nerve (GSPN) and middle fossa floor to the trigeminal porus and petrous ridge, defining Dolenc–Kawase rhomboid boundaries (Figure 1A).

Initially, ICGva after ICA administration and INFRARED800 activation showed no fluorescence from the posterior genu to the trigeminal impression along the ICA course. However, the vein of Labbé and lateral tentorial veins fluoresced within 30 s, confirming perfusion (Figure 1C). Rhomboid drilling proceeded along the petrous ridge from the trigeminal porus to the bisecting line between the arcuate eminence and GSPN, using medial-to-lateral strokes [14,15].

The cochlear safety line (CSL) was delineated, but the absence of a clear ICA landmark prevented defining additional lines. After thinning the anterolateral premeatal triangle by 3 mm (pre-dissection carotid canal roof thickness: 11 mm), a second ICG bolus was administered. This time, fluorescence was emitted from the horizontal petrous ICA, medial to the GSPN, through the thinned bone. The cochlear line (CL) and carotid-cochlear distance (CCD) were successfully measured using the fluorescent medial ICA margin. The posteromedial rhomboid was then drilled, completing the anterior transpetrosal approach. The tentorium was incised and retracted, the carotid canal skeletonized for descriptive purposes, and a partial posterior petrosectomy was performed. Finally, ICGva revealed fluorescence of the intrapetrous ICA from the posterior genu to the petrolingual ligament, as well as the sigmoid sinus, vein of Labbé, vertebrobasilar complex, and ipsilateral anterior inferior cerebellar artery (Figure 1B,D).

### 3.2. Surgical Application

A 41-year-old female presented with progressive short-term memory loss, confusion, fainting episodes, and tinnitus over 12 months. MRI revealed a 2.5 × 3.5 cm left-sided extra-axial petro-tentorial lesion consistent with meningioma, causing brainstem compression (Figure 2). Despite a neurological exam that was intact, documented tumor growth during follow-up prompted surgical intervention. Preoperative imaging identified a 3.5 mm bone layer covering the petrous carotid canal. The patient underwent a Modified Dolenc–Kawase approach with pretemporal-transcavernous and anterior transpetrosal techniques, as outlined in the cadaveric simulation and described previously [13,16].

Following the steps described in the cadaveric simulation, the anterior transpetrosal approach (ATPA) proceeded in a medial-to-lateral direction to reach the transition fold between the IAC roof and the anterior wall first. A 12.5 mg ICG bolus was then administered, and intraoperative ICG videoangiography (ICGva) with NIR-white light overlay visualized the horizontal ICA segment within 20 s (Figure 3A–C), guiding the definition of the carotido-cochlear distance (7 mm from the fallopian hiatus) and the cochlear line (CL).

Based on the definition of the landmarks mentioned above, the ATPA was optimized and completed, followed by an intradural transtentorial approach to resect the petro-clivo-tentorial lesion (Figure 4).

The procedure was uneventful, and the postoperative MRI confirmed the gross total resection of the lesion (WHO grade 1 meningioma). The patient was discharged on postoperative day four without morbidity and returned to work within two months.

## 4. Discussion

This study demonstrates the utility of transosseous ICG video angiography (ICGva) with NIR-white light overlay in delineating the horizontal segment of the intrapetrous ICA during anterior transpetrosal approaches. By providing a real-time surgical landmark, this technique facilitates the definition of the posteromedial triangle boundaries. To our knowledge, this is the first application of ICGva for visualizing the petrous ICA in skull base procedures. At the same time, previous studies had primarily employed ICGva to distinguish tumors from healthy brains or dura mater in endoscopic and posterior fossa surgeries [17,18,19,20].

The anterior transpetrosal approach (ATPA) offers tailored access to the upper and ventrolateral brainstem with minimal retraction. Yet, studies on optimizing posteromedial rhomboid boundaries remain limited despite decades of use. Borghei-Razavi et al. reviewed ATPAs for petroclival meningiomas, finding the anterior drilling limit consistently at least 2 mm (average 5 mm) from the horizontal petrous ICA on postoperative high-resolution CT. They correlated the extent of anterolateral drilling with tumor diameter, highlighting the increased risk of petrous ICA injury during MDKR drilling in larger tumors, while average distances to the IAC and semicircular canals, but not the cochlea (not evaluated), were considered safe [21]. Maximizing the lateral extension of MDK rhomboid drilling requires precise evaluation of the carotid-cochlear spatial relationship. The cochlea, located beneath the middle fossa floor, sits at an angle between the labyrinthine facial nerve and GSPN (apical turn), posteriorly defining the IAC fundus. It is positioned postero-laterally to the geniculate ganglion and is separated by ~2 mm of bone from the horizontal petrous ICA, which lies antero-inferiorly near the basal turn. Preoperative assessment of this relationship is challenging due to the non-coplanar orientation of the posterior ICA genu and cochlea, requiring specific CT reconstructions. Standard axial and coronal scans often fail to capture both structures simultaneously, while millimetric distances complicate intraoperative navigation use, given its intrinsic spatial tolerance [11,22,23].

### 4.1. Sequential Landmarks Adoption

Glasscock’s triangle was among the first intraoperative landmarks described for cochlear localization in middle fossa approaches, defined by the foramen ovale, posterior border of V3, and cochlear apex. Later, Tanriover et al. refined it using the arcuate eminence as the posterior corner, with V3 and GSPN. However, variability in size, shape, and the inconsistency of the arcuate eminence limit its reliability [24,25]. Conversely, the premeatal triangle—bounded by the medial IAC, geniculate Ganglion, and horizontal ICA—consistently locates the cochlea within its lateral half [13,22,23]. Although anatomically solid, it requires carotid canal exposure or reliance on averaged cadaveric measures, which is rarely needed or advisable.

The cochlear safety line (CSL) was recently introduced as a landmark to protect the cochlea during ATPAs. By connecting the lateral rim of the foramen ovale perpendicularly to the upper dural transition fold (UDTF) between the roof and anterior IAC wall, the CSL delineates a medial area for extensive drilling, with a 0.8–4.6 mm lateral margin from the cochlea [10]. In our cadaveric simulation and clinical case, the CSL was successfully implemented early during medial petrous apex drilling, proving useful for mid-posterior drilling of the posteromedial rhomboid. However, it does not provide spatial information about the cochlea-ICA relationship, forcing reliance on the GSPN trajectory, which may not align with the carotid canal orientation [26].

To address this limitation, ICGva allowed us to implement the cochlear line (CL) and carotido-cochlear distance. After GSPN dissection, ICGva reliably identified the carotid canal and the ICA-GSPN intersection. From the latter, a perpendicular line to the UDTF defined the CL. Cadaveric studies report the CL as a safety margin of ~2 mm from the cochlear medial boundary, though its intraoperative use is limited without direct ICA visualization. In our case, combining the CSL and CL intraoperatively enhances spatial orientation, ensuring accurate anterior transpetrosal drilling toward the rhomboid anterolateral margins.

In addition, during the same ICGva, a line from the geniculate ganglion to the medial ICA margin was defined. A 7 mm distance from the ganglion along this line limited the carotid-cochlear safety margin to locate the cochlear basal turn rim. The bony thickness between this point and the carotid canal (averaging 1.5–2 mm, 2.4 mm in our cadaveric specimen) was drilled as necessary, optimizing the anterolateral rhomboid corner to improve maneuverability [12]. Further cadaveric dissection confirmed the utility of ICGva in reliably locating the carotid canal and tracing its trajectory posteriorly to the genu for complete skeletonization. The definition of cochlear-ICA spatial interrelation by triangulating three anatomical landmarks under ICGva visualization was straightforward, carried out in real time during the videoangiography, and helped maintain spatial orientation in a narrow operative field without increasing surgical duration or requiring interruptions from petrous bone drilling for repeated landmarks verification.

### 4.2. Applications

In clinical scenarios, this technique might help optimize the anterior petrosectomy rim, enhancing the anteroposterior and lateromedial motion range toward the ventral brainstem while avoiding breaches within the petrous carotid dural membrane and ICA injury (Figure 4) [27]. In our surgical case, an additional ICGva was unnecessary in later ATPA stages due to the tumor’s limited coronal extension. However, this technique could support further bony expansion with stepwise ICA monitoring via ICGva or a Doppler probe for larger tumors.

Moreover, based on the preliminary findings obtained during our cadaveric trial and prior reports, intraoperative ICGva with new digital overlay modalities may address additional challenges in skull base surgery. While not tested in our clinical case, our cadaveric dissection demonstrated the feasibility of using ICGva during ATPA with partial posterior petrosectomy to approximate the location and distance of the sigmoid sinus [28,29]. During temporal bone dissection in our specimen, sigmoid sinus fluorescence became visible after drilling the posterior petrous ridge and upper lip of the posterior petrous wall, delineating the sinus course and guiding orientation in the presigmoid space (Figure 1D). This application could prove valuable in partial posterior petrosectomies during extended middle fossa approaches or as an early landmark in presigmoid approaches, particularly in temporal bone malignancies with anatomical distortion or variants.

Additionally, ICGva might offer real-time feedback on venous patency. While previous NIR visualization modalities (e.g., FLOW800) lacked straightforward white-light correlation, limiting routine use, the NIR-white light overlay allowed for more precise anatomical integration [30]. In fact, it might also be implemented for detecting intraoperative dural sinus occlusions and Labbe’s vein patency (Figure 1C,D) [28,31,32]. Periodic monitoring of venous patency with ICGva during lengthy skull base procedures could serve as a cost-effective strategy to mitigate retraction-related complications, such as sinus occlusion or cerebral venous thrombosis. Indeed, transdural fluorescence from the Labbe’s vein was easily identified in our cadaveric dissection and clinical case. Nevertheless, these applications have not been explored in our feasibility study and require reproducible evidence from prospective series to support their reliability.

### 4.3. Practical Considerations and Future Directions

In our cadaveric simulation and clinical case, transosseous fluorescence of the pICA was identifiable when the carotid canal roof thickness was less than 8 mm. Although this observation accords with prior anatomical considerations from other contexts (transosseous fluorescence up to 8–10 mm through bone, 2–4 cm in soft tissue), it should be regarded as hypothesis-generating and confirmed prospectively with standardized thickness measurements and blinded overlay-to-imaging localization error [32,33,34,35]. Signal quality is further influenced by tissue composition (bone density and marrow content), surface conditions (extravasation and thermally altered cortical bone), and the pattern of pneumatization. Broadly, pneumatization that thins the mineralized path can facilitate transmission, whereas interposed dry air cells introduce bone–air interfaces that degrade optical coupling and produce patchy contours; fluid “bridging” from irrigation can partially mitigate these effects.

Beyond carotid-canal dehiscence, the technique is likely to be less informative with markedly thick or sclerotic petrous bone, multi-lamellar dry pneumatization, persistent bleeding, or surface carbonization that attenuate NIR transmission, and adjacent hardware that causes glare or shadowing. Patient-specific vascular variants (e.g., aberrant carotid course, high-riding jugular bulb) may alter fluorescence geometry and warrant careful pre-operative scrutiny on high-resolution imaging scans. Standard ICG considerations also apply (hypersensitivity, rare anaphylactoid reactions, pregnancy caution), and practical use presupposes a microscope with integrated NIR–white light overlay, which can represent a relevant financial burden, and team familiarity with its optical and working-distance constraints [36]. Although acquisition and interpretation of ICGva are familiar to many vascular/skull-base teams, additional learning primarily concerns disciplined, sequential landmarking under overlay (CSL → CL → CCD) and error-aware use near curved bony edges where small registration/parallax biases are most likely [37].

Looking ahead, the generalizability of these preliminary findings will depend on prospective, comparative studies that quantify localization error to the pICA and basal cochlear turn against co-registered CT/CTA/MRI, with analyses explicitly stratified by bone thickness. Protocols should also harmonize ICG dose and timing, microscope working distance/zoom, and the timing of overlay acquisition. In parallel, ex vivo and phantom experiments can define detection thresholds across controlled gradients of thickness, density, and pneumatization, and test the impact of surface conditions (e.g., bleeding or thermally altered bone) on signal quality.

### 4.4. Limitations

By design, this report addresses feasibility and potential utility rather than efficacy at the current stage. Although real-time visualisation of the petrous ICA may enhance proximal exposure and keyhole views of the posterior fossa, confirmation will require prospective, comparative studies powered to detect differences in safety, spatial accuracy, exposure, and surgical efficiency.

Several methodological constraints warrant emphasis. Despite being predictive of what is encountered in the surgical scenario, embalming and artificial perfusion in cadaveric specimens can alter optical properties (i.e., altered washout, lack of blood protein binding) and haemodynamics, potentially over- or under-estimating transosseous fluorescence relative to living tissue [36]. The single clinical case limits broad generalizability to thicker bone, marked pneumatization, prior surgery, or variant anatomical conditions. ICG dosing and injection timing followed standard intraoperative vascular neurosurgery practice; however, these parameters have not been validated for transosseous, extradural skull-base guidance, where the optical path (bone, air cells), target depth, and surgical task differ materially. Consequently, the relationship between administered dose, perceived fluorescence intensity, and spatial fidelity of the overlay in this context remains unproven. A quantitative ground truth was not obtained: we did not compute overlay-to-imaging localisation error for the ICA or basal cochlear turn, protocolise micro-Doppler or navigation checkpoints, or assess inter-/intra-rater reliability for landmark placement at this stage.

In addition, optical and overlay caveats are intrinsic to the modality: microscope-integrated NIR ICG with white-light digital overlay registers two imaging channels; minor registration/parallax errors may occur with changes in zoom, focus, working distance, or oblique viewing angles (trajectories of the imaging lens < 60° affect fluorescence intensity acquisition in ex vivo studies). Fluorescence is surface-weighted with limited penetration and is attenuated by cortical bone, air cells, blood film, and drill char; the fluorescent contour should not be construed as a sharply defined anatomical boundary, particularly at the edges of the region of interest where spatial bias is most likely. Motion, irrigation, auto-exposure, and blooming can further blur margins. In practice, transosseous visibility depends on local bone thickness, and construction of the cochlear line and carotido-cochlear distance remains partly operator-dependent (e.g., variability in the GSPN course, fallopian hiatus, and upper dural transition fold).

Finally, ICG is generally safe, yet repeated dosing introduces small risks and may modestly affect its application; conversely, overlays can reduce mode-switch time but risk confirmation bias if over-trusted. To enable a rigorous evaluation, future work should predefine reference standards (co-registered high-resolution CT/CTA/MRI) and quantitative error metrics, incorporate concurrent micro-Doppler/navigation validation, stratify analyses by bone thickness/pneumatization, measure time cost and observer reliability, and compare overlay-assisted versus standard landmark-based drilling in prospective, adequately powered cohorts with prespecified safety and exposure endpoints.

## 5. Conclusions

This technical note describes the intraoperative use of near-infrared indocyanine-green (NIR-ICG) fluorescence with white-light digital overlay during the anterior transpetrosal approach. In a cadaveric model and a single operative case, the method enabled real-time visualisation of the petrous ICA. It also supported the intraoperative placement of cochlear-related reference lines, permitting tailored modification of the MDK rhomboid. The workflow integrated smoothly into the operative sequence, and, in the illustrative case, gross-total resection was achieved uneventfully.

These observations suggest that NIR-ICG overlay can function as a practical adjunct to conventional anatomical landmarking and standard intraoperative tools when drilling near the ICA and cochlea. The present experience is preliminary and descriptive; the technique should be regarded as complementary rather than determinative in surgical decision-making.

These findings are hypothesis-generating and motivate broader evaluation with quantitative metrics and comparative designs to define clinical impact and clarify indications. Until such data are available, selective use of NIR-ICG overlay may be considered in cases where additional, real-time vascular orientation is desirable within a narrow operative corridor.

## 6. Patents

The perfusion system used in the cadaveric experiment is pending patent registration.

## Figures and Tables

**Figure 1 jcm-14-06954-f001:**
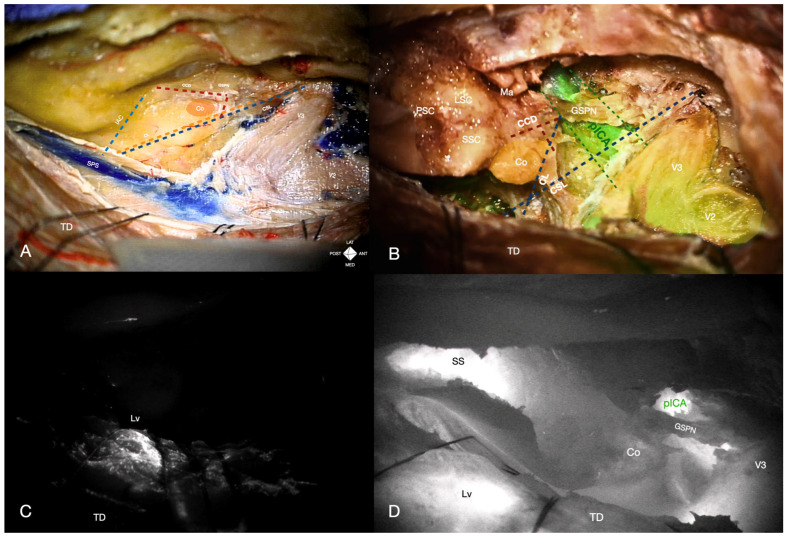
Cadaveric specimen perfused with indocyanine green (ICG), viewed under near-infrared fluorescence and white-light digital overlay. (**A**) Middle cranial floor exposure after pretemporal interdural dissection in an injected specimen. Coloured dotted lines indicate key anatomical references: the cochlear line (CL, orange dotted line), the carotido-cochlear distance (CCD, red dotted line), the cochlear safety line (CSL, blue dotted line), and the internal acoustic canal (IAC, blue dotted line). (**B**) Middle fossa floor dissection and drilling through ATPA and partial posterior petrosectomy. The CL (blue dotted line), CCD (red dotted line), CSL (blue dotted line), and GSPN trajectory are similarly illustrated, highlighting their relationship to the petrous ICA (NIR-white light overlay after ICG perfusion) and adjacent neurovascular structures. (**C**) Intraoperative near-infrared (IR800) ICG videoangiography illustrating the fluorescence of the vein of Labbé. (**D**) IR800 ICG visualization of the petrous segment of the internal carotid artery (pICA), the vein of Labbé (Lv) through the temporal dura (TD), and the sigmoid sinus (SS) after ATPA and partial posterior petrosectomy. CCD, carotido-cochlear distance; CL, cochlear line; Co, cochlea; CSL, cochlear safety line; Ga, Gasserian ganglion; GSPN, greater superficial petrosal nerve; IAC, internal acoustic canal; Lv, Labbé’s vein; LSC, lateral semicircular canal; Ma, malleus; NIR, near infrared; pICA, petrous internal carotid artery; SPS, superior petrosal sinus; PSC, posterior semicircular canal; SS, sigmoid sinus; SSC, superior semicircular canal; TD, temporal dura; V2, maxillary nerve; V3, mandibular nerve.

**Figure 2 jcm-14-06954-f002:**
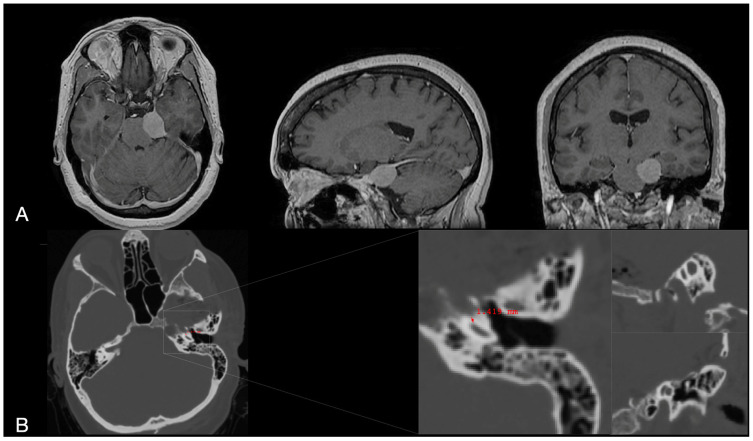
(**A**) Preoperative T1-weighted MRI sequences (axial, sagittal, and coronal planes) demonstrating a left petro-clivo-tentorial meningioma exerting significant mass effect on the brainstem. (**B**) Postoperative high-resolution CT scans following the anterior transpetrosal approach (ATPA) highlight the drilled petrous bone corridor. The magnified views (axial, sagittal, coronal) illustrate the proximity of the drilling volume to the basal cochlear turn and the petrous carotid canal (1.4 mm).

**Figure 3 jcm-14-06954-f003:**
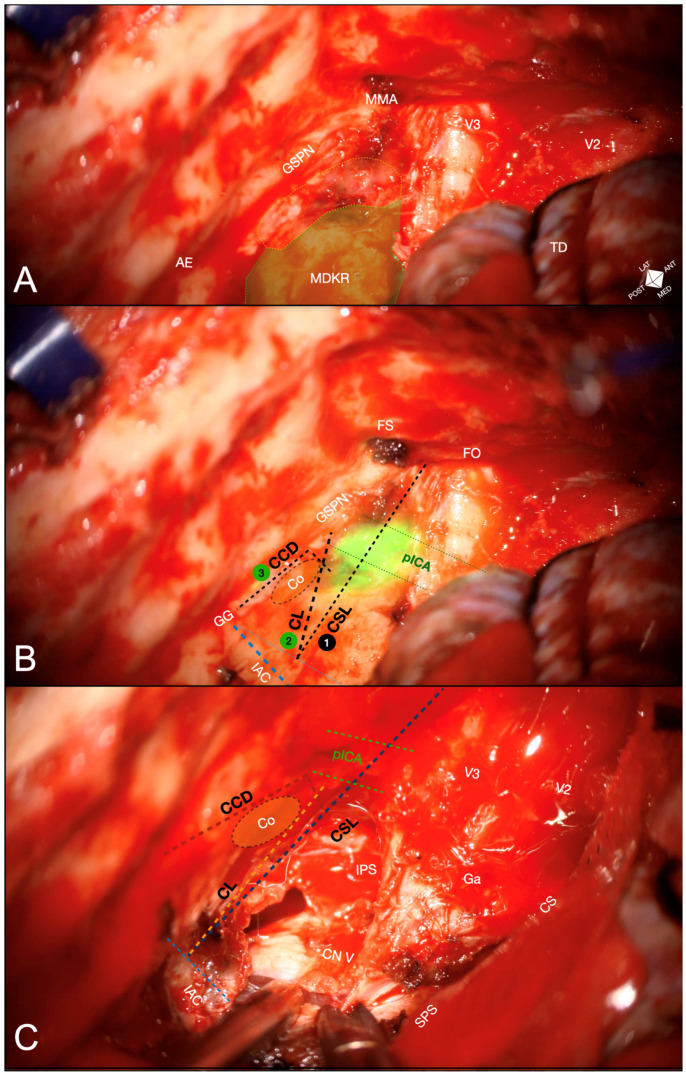
Intraoperative images depicting the ATPA performed for a left petro-clivo-tentorial meningioma. (**A**) Pre-ICGva exposure. Extradural exposure of the middle fossa floor with delineation of the modified Dolenc–Kawase rhomboid (MDKR, green dotted area). The anterolateral rhomboid surface (yellow dotted area) is potentially accessible but offers limited superficial landmarks, carrying risk to the cochlea and internal carotid artery. Extradural reference points (e.g., AE, GSPN, FO/FR/FS, petrous ridge/porus trigeminus) can be identified only; the cochlear safety line (CSL) can be planned from the lateral rim of FO perpendicular to the UDTF at this stage. (**B**) ICGva overlay. After a 12.5 mg ICG bolus, microscope-integrated ICG videoangiography (ICGva) with white-light digital overlay visualizes the petrous carotid (pICA). Landmarks are defined in order: 1 = cochlear safety line (CSL), drawn from the lateral FO perpendicular to the UDTF; 2 = cochlear line (CL), projected perpendicularly from the fluorescent pICA–GSPN intersection to the UDTF; 3 = carotido-cochlear distance (CCD), measured from the fluorescent medial pICA margin toward the fallopian hiatus/geniculate ganglion. This 1 → 2 → 3 sequence helps define the ICA course and cochlear position, refining MDKR boundaries for tailored drilling on its lateral aspect. (**C**) Post-ATPA evaluation. Following completion of the ATPA, the microsurgical view during tumour resection confirms landmark concordance, preservation of carotid-canal boundaries, and avoidance of cochlear violation. AE, arcuate eminence; CCD, carotido-cochlear distance; CL, cochlear line; Co, cochlea; CS, cavernous sinus; CSL, cochlear safety line; FO, foramen ovale; FR, foramen rotundum; FS, foramen spinosum; Ga, Gasserian ganglion; GSPN, greater superficial petrosal nerve; IAC, internal acoustic canal; ICGva, indocyanine-green videoangiography; MDKR, modified Dolenc–Kawase rhomboid; MMA, middle meningeal artery; pICA, petrous internal carotid artery; SPS, superior petrosal sinus; TD, temporal dura; UDTF, upper dural transition fold; V2, maxillary division of trigeminal nerve; V3, mandibular division of trigeminal nerve.

**Figure 4 jcm-14-06954-f004:**
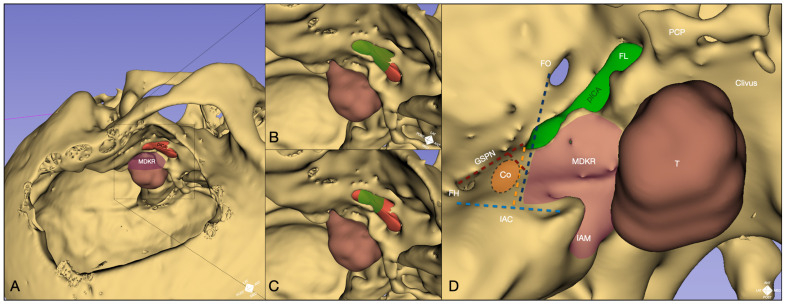
A three-dimensional reconstructed model integrating postoperative CT scans with preoperative MRI data. (**A**) The ATPA corridor through the modified Dolenc–Kawase rhomboid (MDKR) is shown in relation to the skull base. (**B**) The cadaveric experiment confirms ICG fluorescence from the petrous ICA (pICA), establishing a robust anatomical reference. (**C**) Similar fluorescence patterns in the operative scenario validate the translational utility of this technique. The difference in fluorescence visualization must be interpreted considering different dissection exposures in the two conditions, with further drilling above the carotid canal and trigeminal rerouting in the specimen, a maneuver not needed in the surgical scenario. (**D**) A schematic overlay synthesizes these elements, illustrating how near-infrared ICG fluorescence coupled with white-light viewing and carefully defined anatomical landmarks collectively enhance the surgeon’s ability to locate, safeguard, and optimize drilling around the ICA and cochlea (Co) during ATPA to address posterior fossa tumors. CCD: carotido-cochlear distance (red); CL: cochlear line (orange); Co: cochlea; CSL: cochlear safety line (blue); FH: fallopian hiatus; FL: foramen lacerum; FO: foramen ovale; FR: foramen rotundum; FS: foramen spinosum; Ga: Gasserian ganglion; GSPN: greater superficial petrosal nerve; IAC: internal acoustic canal (light blue); IAM: internal acoustic meatus; MDKR: modified Dolenc–Kawase rhomboid; MMA: middle meningeal artery; PCP: posterior clinoid process; pICA: petrous internal carotid artery; SPS: superior petrosal sinus; T: tumor; TD: temporal dura; V2: maxillary nerve; V3: mandibular nerve.

## Data Availability

The original contributions presented in this study are included in the article. Further inquiries can be directed to the corresponding author.

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
