# Peer review of "NIR Indocyanine–White Light Overlay Visualization for Neuro-Oto-Vascular Preservation During Anterior Transpetrosal Approaches: A Technical Note"

_jcm, 2025, doi:10.3390/jcm14196954_

Round 1

Reviewer 1 Report

Comments and Suggestions for Authors

The authors present an interesting and well-prepared technical note introducing the use of near-infrared indocyanine green (NIR-ICG) with white-light overlay as an adjunct in the anterior transpetrosal approach. I found the manuscript both innovative and clinically relevant. The combination of cadaveric simulation and application in a patient provides a compelling demonstration of feasibility, and the intraoperative images are excellent and very helpful in illustrating the workflow.

The introduction provides a solid background, but I felt that it could be slightly sharpened to emphasize the specific clinical challenge this method addresses—namely, the risks to the ICA and cochlea during drilling in this region, and the limitations of relying solely on traditional anatomical landmarks. This would set the stage more clearly for why this technique is needed.

The methods are appropriately described, and the results are well presented. My main reservation is that the conclusions are somewhat stronger than the data currently support. With only one cadaver and one patient included, the findings are preliminary and should be framed accordingly. I think the impact of this report will actually be greater if the authors explicitly acknowledge the limitations of such early-stage work and present the technique as a promising adjunct that merits further study, rather than implying that it already improves outcomes. A brief limitations paragraph would be helpful in this regard.

I also found that while the figures are excellent, some of them feel dense in labeling and might benefit from a bit more simplicity so that the essential landmarks and findings are more immediately clear. The English is overall adequate, but in several places the grammar and sentence flow could be polished to improve readability. A light language edit would make the technical descriptions easier for readers to follow.

In summary, I believe this is a valuable and timely contribution. It introduces a potentially useful adjunct to improve safety in a difficult skull base approach and will be of interest to skull base surgeons. Some refinement of the framing, a more cautious conclusion, a short limitations statement, and modest language polishing is needed.

Comments on the Quality of English Language

The English is generally understandable but could be polished for grammar, flow, and clarity. The writing is not poor, but a light professional edit would strengthen the readability and improve the precision of the technical descriptions.

Author Response

Reviewer #1

1) Sharpen the Introduction to foreground the clinical problem (risk to ICA/cochlea; limits of traditional landmarks).

Response: We tightened the final paragraphs of the Introduction to explicitly frame the unmet need: drilling near the petrous ICA and cochlea remains dependent on broad bony landmarks and subjective orientation, which risks localization error. We now state why an extradural, real-time vascular cue could complement existing landmarks and modernize MDK drilling boundaries. Revised in Introduction, final two paragraphs.

2) Temper conclusions; emphasize preliminary nature; add a brief limitations statement.

Response: We substantially revised both the Discussion and Conclusions to reflect feasibility rather than efficacy and to avoid overstating impact. We added an explicit Limitations subsection (Section 4.4) acknowledging the single cadaver and single patient, lack of quantitative ground truth, and overlay/optical caveats; Conclusions now present the technique as a practical adjunct that warrants prospective validation rather than a proven outcome-improving method. Revised Sections 4.4 and 5. We apologize for the previous overstatement.

3) Figures are dense; simplify labeling so essential landmarks stand out.

Response: We simplified captions and made the labeling logic explicit, particularly for Figure 3, where panel B now shows a sequential 1→2→3 definition (CSL → CL → CCD) under ICGva to ease interpretation of our methodology. Figure 4 was edited to clarify how cadaveric and clinical fluorescence views translate to the schematic, and to avoid acronym collision (e.g., spelling out “clivus” where “CL” also denotes cochlear line). Revised Figure 3 and Figure 4 captions and keys.

4) English language polish.

Response: We performed a light language edit for concision, flow, and technical precision across the manuscript. See tracked changes.

5) Abbreviation consistency.

Response: We harmonized abbreviations across text/figures and expanded the Abbreviations list to include all items appearing in figure keys. We also resolved potential collisions (e.g., CL = cochlear line; “clivus” spelled out in captions). Updated Abbreviations section and figure captions.

Reviewer 2 Report

Comments and Suggestions for Authors

The authors present a novel and intriguing technical note on the use of near-infrared indocyanine green (NIR-ICG) with a white-light overlay to visualize the petrous internal carotid artery (ICA) during anterior transpetrosal approaches (ATPA). The manuscript is well-written, the figures are of high quality, and the described technique addresses a significant challenge in skull base surgery—namely, the safe and precise delineation of neurovascular structures. The integration of a cadaveric simulation with a clinical case provides a strong proof-of-concept for this innovative application of ICG videoangiography. The discussion effectively situates this technique within the context of existing anatomical landmarks and surgical approaches, highlighting its potential to refine the drilling of the modified Dolenc-Kawase rhomboid. This work represents a potentially valuable contribution to the neurosurgical armamentarium for complex skull base procedures.

The primary limitation of this study is its research design, which is based on a single cadaveric specimen and a single clinical case. While appropriate for a "technical note," the conclusions drawn from this limited dataset are overly broad and must be tempered. The assertion that this approach could "enhance intraoperative safety and improve exposure" is speculative and not sufficiently supported by the evidence presented. To strengthen the manuscript, the authors should significantly revise their conclusions to reflect the preliminary nature of their findings. I would recommend explicitly discussing the limitations of an n=1 study in the main body of the discussion, rather than briefly mentioning it at the end. Furthermore, a more in-depth analysis of potential challenges would be beneficial. For example, what is the minimum bone thickness through which fluorescence can be reliably detected? Are there other contraindications besides carotid canal dehiscence? What is the anticipated learning curve for surgeons adopting this technique? Addressing these points would provide a more balanced and critical analysis of the technique's true utility.

Finally, while the manuscript is generally clear, a few minor points could enhance its impact. The authors should ensure consistency in the abbreviations used across figures and text. The discussion regarding the potential application for monitoring dural venous patency is interesting but feels somewhat tangential; this section could be condensed to maintain focus on the primary objective of ICA localization. In closing, this is a well-conceived and clearly presented technical innovation with significant promise. By carefully rephrasing the conclusions to avoid over-generalization and by expanding the discussion on limitations and potential pitfalls, the authors will produce a stronger and more impactful manuscript that accurately reflects the current state of their important work.

Author Response

Reviewer #2

1) Temper conclusions; avoid over-generalization.

Response: We reframed conclusions as hypothesis-generating and removed language implying demonstrated safety or exposure gains. The method is presented as a promising adjunct that integrates readily but requires prospective, comparative evaluation before routine adoption. Revised Section 5; mirrored in Section 4.4 (Limitations).

2) Discuss limitations in the main body (not only at the end).

Response: We integrated limitations directly within Discussion (Sections 4.1–4.3) and then consolidated them in Section 4.4 (Limitations), covering: study design (feasibility only), lack of quantitative overlay-to-imaging error, unvalidated dosing/timing for transosseous guidance, overlay registration/parallax considerations, and operator dependence of specific landmarks. Revised Sections 4.1–4.4.

3) Provide deeper analysis of practical challenges:

Response: We added Section 4.3 (Practical considerations and future directions) to address these explicitly. We report our observed conditions for transosseous pICA fluorescence in the cadaver and the clinical case, and we frame a pragmatic thickness threshold as hypothesis-generating that requires prospective confirmation with standardized measurements and blinded localization error analysis. We also list conditions likely to impede signal (markedly thick/sclerotic bone, multi-lamellar dry pneumatisation, persistent bleeding or surface carbonization, adjacent hardware glare) and reiterate standard ICG safety considerations.Finally we outline the specific incremental skills (disciplined sequential landmarking under overlay—CSL → CL → CCD—and error-aware interpretation near curved bony edges) and suggest verification-rich early use while evidence matures. Added/expanded in Section 4.3; cross-referenced in 4.4.

4) Ensure consistency of abbreviations across figures and text.

Response: Addressed comprehensively; see also response to Reviewer #1. Updated figure keys and Abbreviations list.

5) Condense tangential discussion on dural venous patency.

Response: We retained the concept as an auxiliary observation but condensed and clearly flagged it as outside the scope of the present feasibility demonstration for ICA localization, noting that prospective data would be required for reliability. Revised Section 4.2 (Applications).

Round 2

Reviewer 1 Report

Comments and Suggestions for Authors

The authors have revised the manuscript substantially and addressed the major concerns from the previous review. The introduction now highlights the specific clinical challenges more clearly, and the conclusions have been appropriately tempered to reflect feasibility rather than efficacy. The addition of a dedicated limitations section strengthens the balance of the paper.

Figures and captions have been significantly improved, with labeling now easier to follow and schematic connections clearer. The intraoperative and cadaveric images remain a strong asset. The English language is overall clearer and more polished than before; however, a final professional edit would still help smooth out minor phrasing.

Overall, this is now a well-prepared and informative technical note. It convincingly demonstrates the feasibility of real-time NIR-ICG overlay for anterior transpetrosal approaches and positions the work as a valuable adjunct, warranting further study. I recommend acceptance after only minor language adjustments.

Comments on the Quality of English Language

The English is understandable, but several sentences could still benefit from light professional editing for grammar and flow. The technical content is clear, and the language issues are minor.